# A Chinese Classical Prescription Chaihu Shugan Powder in Treatment of Post-Stroke Depression: An Overview

**DOI:** 10.3390/medicina59010055

**Published:** 2022-12-27

**Authors:** Zu Gao, Yuan Wang, Huayun Yu

**Affiliations:** 1College of Traditional Chinese Medicine, Shandong University of Traditional Chinese Medicine, Jinan 250355, China; 2Shandong Co-Innovation Center of Classic TCM Formula, Shandong University of Traditional Chinese Medicine, Jinan 250355, China

**Keywords:** post-stroke depression, Chaihu Shugan powder, clinical study, pharmacology, traditional Chinese medicine

## Abstract

Post-stroke depression (PSD) is the most common mental health problem after a stroke with an incidence of up to 33%. PSD has a negative impact on the rehabilitation and recovery of motor and cognitive dysfunction after a stroke and significantly increases the chance of the recurrence of neurovascular events. At present, medication is the preferred method of coping with PSD. Modern medicine is still unclear regarding the pathogenesis of PSD, with clinical drug treatment mostly using antidepressants, such as selective serotonin reuptake inhibitor (SSRIs) and serotonin–norepinephrine reuptake inhibitors (SNRIs). However, a high proportion of patients fail to show an adequate antidepressant response and have adverse reactions after taking antidepressants. In recent years, as the advantages of traditional Chinese medicine (TCM) in clinical treatment continue to emerge, Chinese herbal and TCM formulae have begun to enter the awareness of Chinese scholars and even scholars around the world. As a classic formula with a history of more than 400 years, Chaihu Shugan powder (CHSG) has great advantages in the clinical treatment of PSD. Based on existing clinical and experimental studies, this article comprehensively analyzes clinical cases, mechanisms of action, and drug and chemical effects of CHSG in the treatment of PSD in order to provide more clinical experience and experimental theoretical support for CHSG in the treatment of PSD.

## 1. Introduction

Post-stroke depression (PSD) is a common psychological disorder after a stroke, which involves a series of affective disorder syndromes characterized by low mood and the loss of interest in daily life alongside stroke symptoms [1]. The incidence of PSD is as high as 31% within 5 years after a stroke [2]. PSD not only affects the rehabilitation of neurological function and reduces the quality of life of patients but also increases the disability rate and mortality rate. Some studies have found that the mortality rate of PSD is about 1.28–1.75 times that of stroke patients without affective disorders [3]. According to the theory of traditional Chinese medicine (TCM), PSD is a “Yu Zheng”, occurring after the “stroke” [4]. Chinese herbal and TCM formulae are effective in the treatment of PSD and can significantly improve patients’ depression and neurological deficit symptoms [5]. At the same time, not only is Chinese herbal medicine multi-target and multi-channel but its lesser adverse effects and lower price are more likely to be favored by patients.

As a classical formula in TCM, Chaihu Shugan powder (CHSG) was first recorded in the Jingyue Complete Library of the Ming Dynasty China and has been used for the clinical treatment of emotional diseases for nearly four centuries in China and the Far East. It consists of *Bupleurum chinese* DC. (Radix Bupleuri), *Cyperus rotundus* L. (Rhizoma Cyperi), *Ligusticum chuanxiong* Hort. (Rhizoma Chuanxiong), *Citrus reticulata* Blanco (Pericarpium Citri Reticulatae), *Citrus aurantium* L. (Fructus Aurantii), *Paeonia lactiflora* Pall. (Radix Paeoniae Alba), and *Glycyrrhiza uralensis* Fisch. (Radix Glycyrrhizae) (Table 1). The combination of seven Chinese herbs has a good effect regarding the regulation of emotions and can be used to relieve flank pain, emotional depression, and other symptoms.

In order to comprehensively and thoroughly explore the mechanism of action of CHSG in the treatment of PSD, this paper will combine clinical research and basic research; summarize the clinical case observation, molecular mechanism of action, and pharmacological effects of Chinese medicine; and provide theoretical support for clinical use and experimental exploration.

## 2. Clinical Application of CHSG in the Treatment of PSD

### 2.1. Clinical Case Research of CHSG for PSD

In modern TCM clinical practice, regulating qi to reduce depression is the basic treatment principle of PSD. CHSG can significantly improve the symptoms of PSD, which has important clinical significance for rehabilitation after a stroke [6]. Professor Sheng believes that the core pathogenesis of PSD is liver loss and drainage. The pathological nature of PSD is mostly empirical in the early stages and displays mixed deficiency and excess syndrome in the later stages. A patient with PSD was treated with an oral CHSG decoction combined with repetitive transcranial magnetic stimulation. After 45 days of continuous treatment, the patient’s symptoms, such as loss of interest and depressed mood, were significantly improved. After treatment, the patient’s Hamilton Depression Scale (HAMD) score was 13, the Hamilton Anxiety Scale (HAMA) score was 9, and the self-rating Depression scale (SDS) score was 8. Compared with the initial diagnosis, the treatment effect was significant [7].

According to the unique pathological basis of PSD, Professor He divides PSD into three types: liver stagnation and spleen deficiency type, phlegm and blood stasis mutual obstruction type, and Yin deficiency and fire hyperactivity type [8]. Professor He used an oral CHSG decoction both with and without the formula to treat PSD patients with liver stagnation and spleen deficiency. After 35 days of continuous treatment, the patients’ speech levels increased significantly, their mental state improved significantly, their family members’ appeals improved significantly, and they gradually took the initiative to communicate with others. Their lack of taste was significantly reduced, abdominal distension and wan oppression essentially disappeared, their diets improved significantly, and the symptoms that the patients presented at their first visits were significantly relieved.

Professor Chen [9] initially treats PSD through the liver and believes that the stagnation of the liver qi is the key to the various syndrome types of this disease. He takes “soothing the liver, regulating qi and relieving depression” as the treatment principle. He utilizes an oral CHSG decoction plus or minus formula to treat PSD, and the curative effect is remarkable. Taking one case as an example, before the treatment, the patient was depressed, unhappy, and complained of chest fullness, hypochondrium fullness, dizziness, sticky mouth, bitter mouth, and other symptoms. Their HAMD score was 16. After 15 days, the patient’s family members reported that the patient’s depression and lack of interest in life were significantly improved, their appetite was good, night dreams were significantly reduced, and their stool and urination were normal. Their HAMD score was 11 points after treatment, which was 5 points lower than during their first visit, and the curative effect was obvious.

### 2.2. Clinical Observation of CHSG in Treating PSD

Clinical studies have shown that CHSG can effectively treat PSD, whether it is used alone, CHSG plus or minus formula, or CHSG combined with other drugs and formulas (Table 2). A study [10] reported on the CHSG treatment of liver qi stagnation-type PSD and analyzed its effects on depression, sleep quality, the recovery of neurological impairment, and TCM syndromes before and after treatment. The results showed that the scores of HAMD, Pittsburgh Sleep Quality Index (PSQI), National Institutes of Health Stroke Scale (NIHSS), and TCM syndrome integral scale had all decreased compared with the scores before treatment. In another report [11], 82 PSD patients were divided into two groups. The control group was treated with Deanxit, and the observation group was treated with an oral CHSG decoction and Deanxit. After 2 months of treatment, the HAMD score, the incidence of adverse reactions, and the NIHSS score of the observation group were significantly lower than those of the control group, and the total effective rate of the control group was significantly lower than that of the observation group. Compared with the control group, CHSG combined with Deanxit can effectively relieve depression in the treatment of PSD, and the incidence of adverse reactions is lower, which is worthy of selection in PSD patients. In another clinical observation [12], sixty patients with PSD were randomly divided into an observation group and a control group. The control group was given escitalopram spuronate tablets and the observation group was given an oral CHSG decoction plus formula on the basis of escitalopram spuronate tablets. After 6 weeks of treatment, the HAMD scale score and NIHSS score of the two groups were significantly lower than those before treatment, indicating that either Espresso Escitalopram Oxalate Tablets alone or combined with CHSG treatment are able to effectively improve depressive symptoms. At the same time, after 6 weeks of treatment, the HAMD scale score and serum hs-CRP level of the observation group were better than those of the control group.

CHSG not only directly treats PSD but also reduces the probability of depression in stroke patients [13]. A different study randomly divided 60 stroke patients into two groups. The control group received basic stroke treatment, such as thrombolysis, antiplatelet, anticoagulation, etc., medication. The observation group was given an oral CHSG decoction in addition to the above basic treatment. After 24 weeks of continuous treatment, the number of patients with depression, HAMD score, and NIHSS score in both groups were recorded at weeks 4, 8, 12, and 24. The results showed that different proportions of patients developed PSD despite relevant treatment in both groups after stroke onset. However, the number of patients with PSD, HAMD score, and NIHSS score (10; 7.35 ± 2.87; 4.10 ± 1.39) in the observation group were significantly lower than that of the control group (21; 11.80 ± 4.93; 7.81 ± 3.07), which indicated that, regarding stroke treatment, CHSG can, to a certain extent, prevent the occurrence and development of PSD and promote the rehabilitation of neurological function.

### 2.3. CHSG Improves the Adverse Reactions in the Treatment of PSD

Selective serotonin reuptake inhibitors (SSRIs) are for example fluoxetine and paroxetine, sertraline, fluvoxamine, citalopram, and escitalopram citalopram, while serotonin–norepinephrine uptake inhibitors (SNRIs) include venlafaxine and duloxetine. These two types of antidepressants are the most commonly used in clinical practice, and there are varying degrees of adverse reactions in long-term use [14]. For example, fluoxetine can cause dry mouth, constipation, dizziness, headache, stomach discomfort, diarrhea, panic attacks, tremors, and leukopenia. The main adverse reactions of paroxetine include dry mouth, nausea, anorexia, constipation, headache, tremor, fatigue, and insomnia. Common adverse effects of venlafaxine are fatigue, headache, somnolence, sweating, and nervousness. Compared with the above drugs, CHSG has the advantages of less adverse reactions and good compliance with antidepressant treatment. Wang [15] divided 80 patients into an observation group and a control group, who were treated with an oral CHSG decoction and paroxetine, respectively. After 6 weeks, the total effective rate was 82.5% in the observation group and 85% in the control group. However, during treatment, it was found that there were more adverse reactions in the control group, such as dry mouth, constipation, excitement, insomnia, vertigo, headache, palpitation, tremor, nausea, vomiting, etc., while the adverse reactions in the observation group were less or lighter, such as only mild headaches, fatigue, constipation, sweating, bitter mouth, and so on. Studies suggest that CHSG has the same efficacy as paroxetine, but CHSG has fewer adverse reactions. Chen [16] randomly divided 94 PSD patients into two groups. The observation group was treated with an oral CHSG decoction plus and the control group was treated with fluoxetine. After 4 weeks of treatment, a good response was seen in both groups, and while there was no significant difference between groups (*p* > 0.05), the TESS score in the observation group was significantly lower than that in the control group (*p* < 0.05). The number of patients with nausea, sinus tachycardia, stomach discomfort, dry mouth, and drowsiness were 1, 0, 2, 2, and 1 in the observation group and 14, 5, 15, 18, and 15 in the control group. This suggests that the incidence of adverse reactions in the treatment of PSD with CHSG plus is less than that with fluoxetine.

Clinical studies have found that CHSG can reduce the side effects of combined antidepressants, such as paroxetine and fluoxetine, to a certain extent. In the study by Huang [17], 80 PSD patients were randomly divided into two groups. The control group was treated with fluoxetine and the observation group was treated with an oral CHSG decoction combined with fluoxetine for 4 weeks. The results showed that the total effective rate of the observation group was 95% and the total effective rate of the control group was 80%. In addition, the two groups presented dizziness, taste, nausea, wind, and other adverse reactions to the treatment, but as the incidence of adverse reactions in the observation group was 20%, while in the control group it was 35%, there was a significant difference. CHSG has been shown to have similar effects on other types of depression. For example, in the study by Yang [18], 60 patients with Parkinson’s disease and depressive symptoms were randomly divided into two groups. The control group was treated with paroxetine, and the observation group was treated with an oral CHSG decoction plus or minus paroxetine. After 8 weeks of treatment, the total effective rate of the antidepressant in the observation group was 90%, while that in the control group was 70%. Adverse reactions, such as dry mouth, fatigue, insomnia, and gastrointestinal symptoms, occurred in the early stages of treatment in the two groups. The number of adverse reactions was 13 in the observation group and 26 in the control group, which showed that the efficacy of CHSG plus or minus paroxetine was better than that of paroxetine alone, and the number of adverse reactions was less. In the above clinical observation, compared with the use of conventional antidepressants alone, the combination of CHSG can reduce the incidence of adverse reactions. However, due to the lack of relevant studies, the specific mechanism involved is still unclear and needs further research.

## 3. Mechanism of CHSG in the Treatment of PSD

### 3.1. CHSG Increased the Level of Neurotransmitters in PSD

The neurotransmitter theory is the classic theory of the pathogenesis of depression. This theory posits that the biological basis of PSD is related to the imbalance of the 5-hydroxytryptamine (5-HT), norepinephrine (NE), and dopamine (DA) systems, and depression is caused by the imbalance of the 5-HT, NE, and DA systems. It is thus caused by the insufficient function of the monoamine neurotransmitters NE and 5-HT in the brain [19], and the severity of depression is negatively correlated with the level of monoamine neurotransmitters in the body. After treatment with antidepressant drugs that increase the content of monoamine neurotransmitters in PSD patients, depressive symptoms can be relieved [20]. Wang [21] divided 60 rats into a blank group, a model group, a fluoxetine group, a CHSG medium dose group, and a CHSG high dose group. Except for the blank group, the depression model was established by the intraperitoneal injection of 0.5 mg/kg of RHX for 14 days. After modeling, the fluoxetine group was given fluoxetine solution (9 mL/kg) by gavage, the CHSG group was given CHSG decoction (5 g/kg, 10 g/kg) by gavage, and the other groups were given the same amount of distilled water by gavage. After 14 days of continuous administration, the contents of monoamine neurotransmitters 5-HT, NE, and DA in the hippocampus were measured using ELISA kits. The results showed that compared with the blank group, the hippocampal monoamine neurotransmitters (5-HT, NE, DA) in the model group were significantly decreased (*p* < 0.05), while compared with the model group, the fluoxetine and CHSG groups had significant increases in neurotransmitters (5-HT, NE, DA) (*p* < 0.05). This suggests that CHSG may alleviate depressive symptoms by modulating neurotransmitter levels. Li [22] used an ED723 high performance liquid chromatography electrochemical detector to detect the level of 5-HT and enzyme-linked immunosorbent assay to detect the level of BDNF and found that the combination of CHSG plus or minus formula and transcranial electrical stimulation had a good therapeutic effect on post-stroke depression, which was able to improve the levels of 5-HT and BDNF in PSD patients and reduce their degree of depression and anxiety.

### 3.2. CHSG Protects Hippocampal Neuronal Cells in PSD

The brain-derived neurotrophic factor (BDNF) is expressed in multiple brain regions, such as the cerebral cortex and hippocampus, which have biological functions to maintain synaptic growth and neuronal growth, differentiation, and survival. It is the most widely-distributed neurotrophic factor in the brain [23]. Most of the functions of BDNF in neuronal growth, maturation (differentiation), and maintenance have been attributed to transmission through the tyrosine kinase receptor B (TrkB) [24]. When mature, BDNF released from dendrites binds to and activates the TrkB receptor, and the signal passes through the renin-angiotensin system (Ras) pathway to achieve transduction reactions [25], which protect neurons from the toxic effects of glutamate and activate mitogen-activated protein kinases (MAPK) and the phosphatidylinositol 3-kinase (PI3K) signaling pathway, thereby promoting neural survival and increasing synaptic plasticity and nerve regeneration [26]. BNDF is closely related to the occurrence and development of depression, including PSD [27]. It has been confirmed in a number of studies that the serum BNDF level in patients with PSD is lower than that in patients without depression, and antidepressants can enhance the expression of BDNF in the brain, thereby alleviating depressive symptoms [28,29]. The same conclusion was reached in the study of CHSG in the treatment of PSD. In the study by Hu [30], middle cerebral artery occlusion (MCAO) combined with chronic unpredictable mild stress (CUMS) was used to induce depression-like behavior to establish a PSD rat model. After the intervention of a CHSG decoction by gavage, Western blot was used to detect the expression levels of BDNF and TrkB in the hippocampus of PSD rats. The results showed that compared with the normal group, the expression levels of BDNF and TrkB in the hippocampus of PSD rats decreased, and the expression levels of BDNF and TrkB increased after CHSG intervention. This suggests that the effect of CHSG on PSD can be achieved through the BDNF/TrkB pathway. Yan [31] established a depression model of CUMS rats and used a CHSG decoction alone or a CHSG decoction combined with fluoxetine to intervene in the CUMS rat model. Subsequently, the sucrose preference test (SPT), forced swimming test (FST), open field test (OFT), and Y-maze test were used to evaluate depression-like behavior and cognitive function. Western blot and RT-PCR were used to study the expression of BDNF and BDNF mRNA in the hippocampus and frontal cortex. The results showed that compared with the CUMS group, treatment with CHSG or CHSG combined with fluoxetine could alleviate the depressive symptoms and improve the cognitive function of CUMS rats. BDNF and BDNF mRNA expression levels were also significantly increased in hippocampus and frontal cortex. In addition, Deng [32] also conducted an antidepressant study of CHSG and used immunohistochemistry and reverse transcription–polymerase chain reaction (Rt-PCR) to detect the mRNA expression levels of BDNF and TrkB in the hippocampus, amygdala, and frontal lobe. The results showed that the expression levels of BDNF and BDNF mRNA were significantly decreased in the rat depression model (*p* < 0.05). The expression of BDNF and BDNF mRNA increased after CHSG gavage (*p* < 0.05). This suggests that CHSG may improve the depressive state of the model rats by increasing the mRNA expression of BDNF and TrkB in the hippocampus, amygdala, and frontal lobe.

### 3.3. Protection of PSD Hippocampal Neuronal Cells by CHSG

In the development of depression, hippocampal neuron damage and plasticity disorder are crucial, and antidepressant drugs can play an antidepressant role by reducing hippocampal neuron apoptosis or death [33]. Changes in the structure and function of the CA3 region of the hippocampus can affect the body’s emotion and memory. The decrease or damage of neurons in the CA3 region can lead to emotional abnormalities [34]. Neuronal damage in hippocampal CA3 region is closely related to neuronal apoptosis [35]. Zhang [36] observed the neuronal apoptosis in the hippocampal CA3 region of rats through electron microscopy and flow cytometry. The results of electron microscopy showed that the neurons in the hippocampal CA3 region of depression model rats showed typical apoptotic manifestations, including reduced organelles, the flocculent degeneration of mitochondria, expanded endoplasmic reticulum, the disintegration and degranulation of Nisl bodies, obvious nuclear pyknosis, and unclear synaptic structure in the neuropile area. The results of the flow cytometry showed that the number of apoptotic cells in the hippocampal CA3 region of the model group was significantly higher than that of the normal group. After the intragastric administration of a CHSG decoction, the morphology of hippocampal neurons was significantly improved, the integrity of cell membrane was restored, the nuclear membrane was clear, the Nissl bodies of neurons were clearly visible, the synaptic structure in the neurofelt area was intact, and the number of synaptic vesicles was moderate. The number of apoptotic cells in hippocampal CA3 region was significantly reduced. It has been suggested that CHSG can inhibit the apoptosis of hippocampal neurons in depression model rats, which may be one of the mechanisms of CHSG in the treatment of depression.

B-cell lymphoma-extra-large (Bcl-xL) is an important member of the Bcl-2 family discovered in recent years. It stabilizes the mitochondrial outer membrane by antagonizing the pro-apoptotic proteins of Bcl-2 family (such as Bax and Bcl-xs) or interfering with the assembly of death-inducing signing complex (DISC) and inhibiting cysteinylasparate specific proteinase-8 (caspase-8). It has antiapoptotic effects [37]. Bcl-xs is the reverse regulator of Bcl-xl and has the effect of promoting apoptosis [38]. Fan [39] observed the effect of CHSG decoction on the apoptosis of hippocampal neurons in PSD model rats. The immunohistochemical method was used to detect the expression of Bcl-xs and Bcl-xl proteins in the hippocampus of rats. Compared with the model group, the expression of Bcl-xs proteins in the CHSG group decreased, and the expression of Bcl-xl protein increased. It is speculated that CHSG may regulate the expression of Bcl-xs and Bcl-xl genes to prevent the apoptosis of hippocampal neurons.

Autophagy, also known as type II programmed death, plays an important role in cell proliferation and structural renewal [40]. Microtubule-associated protein light chain 3 (LC3) is a marker of autophagy. During the formation of autophagy, LC-3I will enzymatically dissociate a small fragment of polypeptide and convert to LC-3II. The ratio of LC-3II/LC-3I can be used to determine the level of autophagy. Beclin-1 protein is the coding product of the mammalian autophagy-related gene beclin-1, which participates in the formation of autophagosomes and regulates autophagy activity [41]. Xu [42] used CHSG decoction to intervene in the rat depression model and detected the expression of autophagy protein LC-3 and autophagy gene Beclin-1 protein in hippocampal neurons of rats. Compared with the control group, the LC-3II/LC-3I ratio and Beclin-1 protein expression level of hippocampal cells in the treatment group decreased, suggesting that the antidepressant effect of CHSG may be related to the reduction of autophagy in hippocampal neurons of rats.

### 3.4. CHSG Reduced the Inflammatory Response in PSD

In recent years, the “inflammation hypothesis” of post-stroke depression has also become an important research direction for the pathogenesis of post-stroke depression. The expression levels of peripheral inflammatory cytokines, such as interleukin (IL)-1B, IL-6, IL-18, and tumor necrosis factor (TNF)-α, were significantly increased after the onset of an ischemic stroke [43]. Chronic immune inflammatory response can lead to changes in emotional and cognitive functions related to depression, which may be the pathophysiological basis of the “inflammation hypothesis” [44]. As a stress state, elevated inflammatory cytokines can participate in the occurrence of PSD by increasing the activity of the hypothalamic-pituitary-adrenal (HPA) axis [45]. Studies have shown changes in inflammatory markers after the treatment of PSD. For example, in an animal experiment, serum IL-6 and TNF-α levels and hippocampal NF-κB expression in PSD rats were significantly higher than those in normal rats. After CHSG treatment, the serum TNF-α level and the expression of NF-κB in the hippocampus decreased significantly compared with PSD rats, and the above-mentioned indexes in the high-dose group decreased more significantly, indicating that CHSG has the potential to inhibit neuroinflammation [30]. Another study [46] measured the serum levels of IL-6 and TNF-α in PSD patients and found that the levels of these inflammatory factors were significantly reduced after CHSG treatment. The above studies have shown that inflammation is closely related to post-stroke depression, and CHSG can significantly reduce the inflammatory response of PSD.

In conclusion, CHSG exerts a certain antidepressant effect mainly by increasing the level of neurotransmitters, promoting the secretion of neurotrophic factors, reducing hippocampal neuronal apoptosis, protecting hippocampal neuronal damage, and reducing inflammatory response (Table 3), but the mechanism of action is still unclear and needs further in-depth study.

## 4. Pharmacological Study of Seven Herbs in CHSG

### 4.1. Radix Bupleuri

As the main drug of CSGS, Radix Bupleuri has both evacuation and antipyretic functions, can soothe the liver and relieve depression, can lift Yang qi, and has multiple pharmacological effects, such as being antitumor, antidepression, antifibrosis, and neuroprotection [47]. Researchers extracted and analyzed the chemical components of Radix Bupleuri and found that it mainly contains saponins, volatile oils, flavonoids, polysaccharides, alkynes, and trace elements, with the saponins including saikosaponins a, saikosaponins d, saikosaponins e, and saikosaponins c [48,49,50]. Volatile oils mainly include L-ascorbyl 2, 6-dipalmitate, 2, 4-sebacedienal, alkynyl alcohol and cis, cis-9, and 12-octadecadiene-1-ol [51]. Flavonoids mainly include kaempferol-3-O-α-L-arabinofuranoside and kaempferol-3,7-di-O-α-L-rhamnopyranoside as two kinds of flavonoids [52]. Polysaccharides mainly include L-arabinose, ribose, D-xylose, L-rhamnose, D-glucose, and D-galactose [53].

Saikosaponin is one of the main components of Radix Bupleuri. One study [54] used saikosaponin A to intervene PSD rats, followed by behavioral tests, including OFT, bead-walking test, SPT and FST, and found that saikosaponin A could effectively improve depression-like behaviors in PSD rats. They further evaluated neuronal apoptosis and the expression levels of p-CREB, BDNF, Bcl-2, Bax and Caspase-3 in the hippocampus. The results suggested that saikosaponin A improved depression-like behavior and inhibited neuronal apoptosis in the hippocampus, possibly by increasing the expression of BDNF, p-CREB and Bcl-2 and decreasing the expression levels of Bax and Caspase-3. Other studies have shown that saikosaponins are closely related to the function of the cholinergic nervous system, and the hyperfunction of the cholinergic nervous system can lead to the occurrence of depression [55]. Acetylcholine (ACh) is an important neurotransmitter in the cholinergic nervous system. It is synthesized and released into the synaptic cleft by choline acetyltransferase (ChAT). ACh is often accompanied by the rise of the function of cholinergic nerve disease, and as synaptic cleft ACh can be made of acetylcholinesterase (AChE) degradation, AChE is the key enzyme of the regulated extracellular ACh level [56]. Because ACh is very unstable, its content is often indirectly reflected by ChAT and AChE. In an animal experiment, Zhang [57] used CUMS to prepare a rat depression model and used saikosaponin to intervene in depressed rats. The expression of AChE and ChAT in the hippocampus was detected by immunohistochemical staining. The results showed that compared with the blank control group, the protein expressions of AChE and ChAT in the hippocampus of the model group were significantly increased. After saikosaponin intervention, the expressions of AChE and ChAT in the hippocampus were significantly decreased. These results suggest that saikosaponin may play an antidepressant role by reducing the activity of the cholinergic nervous system.

### 4.2. Rhizoma Cyperi

Rhizoma Cyperi is known for its ability to clear the liver, relieve depression, regulate qi, and relieve menstrual pain. Modern studies have shown that the chemical components of Rhizoma Cyperi mainly include terpenes, flavonoids, alkaloids, sugars, sterols, and other components, which have pharmacological effects such as antitumor, antidepression, anti-inflammatory, antibacterial, antioxidation, and hypoglycemic effects [58]. Volatile oils, including α-vanvanone, vanvanolone, etc., are the main components of Rhizoma Cyperi, with a mass fraction of 0.65–1.4% [59,60]. Flavonoids mainly include quercetin, kaempferol, luteolin, etc. [61], while triterpenoids mainly include dandelion terpene ketone, Sazeraya terpene and damadienol acetate [62].

In their study, Wang [63] used reflux extraction to prepare the 95% ethanol extract of Rhizoma Cyperi by mixing the crude powder of 1 g of Rhizoma Cyperi decoction with 10 mL 95% ethanol (1:10), so that the concentration of the medicinal solution was 1 g/mL (crude drug). After 7 days of continuous gavage (2 g/kg), the mice were subjected to a tail suspension test (TST) and FST. The results showed that the 95% ethanol extract of Rhizoma Cyperi could significantly shorten the tail suspension immobility time and swimming immobility time of the mice, suggesting that the 95% ethanol extract of Rhizoma Cyperi might be the active extract of Rhizoma Cyperi with an antidepressant effect. In another animal experiment [64], the ethyl acetate extract and n-butanol extract of Rhizoma Cyperi were found to have similar potency to the control fluoxetine, both of which shortened swimming immobility time and tail suspension immobility time, and significantly increased the 5-HT and dopamine (DA) levels in the frontal cortex of mice, confirming the significant antidepressant effect of the two extracts on the animal model of “behavioral despair”.

### 4.3. Rhizoma Chuanxiong

Rhizoma Chuanxiong has the functions of activating blood circulation, promoting qi circulation, eliminating wind, and relieving pain, and has a variety of pharmacological effects such as analgesia, anti-inflammation, antioxidant capacity, antitumor, anticoagulation, antidepressant, antiaging, antiatherosclerosis, cell protection, and the improvement of cardiac function [65]. Its chemical composition mainly includes alkaloids, volatile oils, polysaccharides, lactone, organic acids, etc. Of these, alkaloids mainly include ligusticine A, ligusticine B, adenosine, 2′-O-methyladenosine, etc. [66]. Volatile oils mainly include ligustilide, 3-butylphthalide and pinene [67]. Rhizoma Chuanxiong polysaccharide is mainly composed of glucose, galactose, arabinose, and xylose [68].

In the antidepressant study of Rhizoma Chuanxiong, researchers used the Rhizoma Chuanxiong injection in PSD rats, and the results showed that the Rhizoma Chuanxiong injection was able to improve the behavior and neurological function of PSD rats by upregulating the cyclic adenosine monophosphate (cAMP)-cAMP response element binding protein (CREB)-BDNF pathway; protect the nerve cells in the hippocampal CA1 region; and alleviate the impairment of cognitive function in depressed rats [69]. In addition, Wu [70] used Rhizoma Chuanxiong volatile oil to intervene in rats in a depression model to study the behavior and brain levels of dopamine and norepinephrine secretion in rats. Surprisingly, Rhizoma Chuanxiong volatile oil increased the level score and percentage of sugar-water preference in the depression model rats in the absentee field experiment and reduced the swimming immobility time. Another conclusion of the experiment was that Rhizoma Chuanxiong volatile oil significantly increased the DA content in hippocampus, prefrontal, and striatal NE levels in the depression model rats. These results suggest that the antidepressant effect of Rhizoma Chuanxiong volatile oil may be related to the increase in prefrontal and striatal NE content and hippocampal DA content.

### 4.4. Pericarpium Citri Reticulatae

Pericarpium Citri Reticulatae has the effect of regulating qi, strengthening the spleen, drying dampness, and resolving phlegm, and it has various effects such as antibacterial, anti-inflammatory, antioxidant, antitumor, digestive, expectorant, hepatoprotective, hypotensive, and neuroprotective effects [71]. Its main components comprise flavonoids, volatile oil, alkaloids, trace elements, and other substances [72]. Flavonoids mainly include hesperidin, neohesperidin, citrus, dihydronobiletin, and 5-nordihydronobiletin [73]. Volatile oils mainly include limonene, γ-terpineene, β-laurene, and α-terpineol [74]. Its main alkaloid substance is synephrine, also known as p-hydroxyphenylephrine, which is found in high amounts. In addition, Pericarpium Citri Reticulatae also contains a variety of trace elements, such as potassium, sodium, calcium, magnesium, copper, zinc, iron, and strontium [75].

Inflammation, neurotransmitters, and abnormal apoptosis of neuronal cells are the potential pathogenesis of PSD. Pharmacological studies have found that Pericarpium Citri Reticulatae extract has good anti-inflammatory, antioxidation, and good neuroprotective effects. As the main active ingredient of Pericarpium Citri Reticulatae extract, hesperidin can inhibit NF-κB and inflammatory factors, such as IL-1β, IL-6 and TNF-α, indicating that hesperidin has obvious anti-inflammatory potential [76]. In addition to this, the potential neuroprotective molecular mechanisms of hesperidin and its metabolites have been the subject of numerous studies, suggesting that the neuroprotective potential of hesperidin, such flavonoids, is mediated by improving nerve growth factors and endogenous antioxidant defense and reducing neuroinflammation and apoptosis pathways [77].

### 4.5. Fructus Aurantii

Fructus Aurantii has the ability to regulate qi, broaden the middle-jiao, move stagnation, and reduce swelling. Modern pharmacological studies have found that it has the effects of regulating the gastrointestinal tract, antidepression, immune regulation, and so on [78]. Its main chemical components include flavonoids, coumarin, alkaloids, and volatile oils [79]. Of these, flavonoids are the main active ingredients in Fructus Aurantii, including flavonoids, flavonols, isoflavones, dihydroflavonoids, dihydroflavonols, chalcones, anthocyanins, etc. [80]. The main components of volatile oil are limonene, linalool and α-terpineol [81], among which limonene, as the main volatile oil of Fructus Aurantii, is an important active component for its qi-regulating effect [82]. Alkaloids mainly include synephrine, tyramine, and N-methytyramine [83].

In an animal experiment, the ethanol extract of Fructus Fructus Aurantii significantly reduced the levels of corticosterone, increased sugar preference, and reduced the immobility time during the forced swimming test in stressed rats, which revealed the antidepressant effect of fructus. Moreover, its effect on alleviating depressive symptoms may be associated with increased gastric motility in rats, the upregulation of hippocampal GR mRNA and the BDNF mRNA expression in the cortex and hippocampus [84]. Other experiments [85] suggested that the protective effect of Fructus Aurantii on nerve cells may also be related to its antidepressant mechanism. In addition, the inhibition of the hypothalamic-pituitary-adrenal axis hyperactivity, the increase in hippocampal glucocorticoid receptor and corticobrain-derived neurotrophic factor mRNA expression, the increase in neurotransmitter class expression and release, the regulation of gastrointestinal hormones, and the increase in gastrointestinal motility are also potential mechanisms of the antidepressant action of citrus aurantium [86].

### 4.6. Radix Paeoniae Alba

Radix Paeoniae Alba has the ability to replenish blood, regulate menstruation, astringent fluid, stop sweating, reduce liver pain, and inhibit the hyperactivity of liver yang. Modern pharmacological studies have found that white peony has many effects, such as anti-inflammatory, analgesic, hepatoprotective, and antioxidant effects. It is also rich in chemical components, mainly monoterpenes and their glycosides, triterpenoids, and flavonoids [87]. Monoterpenes and their glycosides are the main chemical substances in peony, including paeoniflorin, oxidized paeoniflorin, and benzoyl paeoniflorin [88]. Triterpenoids in Radix Paeoniae Alba include oleanolic acid, ivy sapogenin, 3β-hydroxyoleane-12-ene-28-acid, and paeoniflorone [89].

Radix Paeoniae Alba has a significant antidepressant effect. In one study, paeoniflorin was found to promote the recovery of neurological function and improve the depression state of PSD rats and increase the expression levels of BDNF and pCREB protein in hippocampal neurons of PSD rats. The preventive and therapeutic effect of paeoniflorin on PSD rats was similar to fluoxetine [90]. Other studies have shown that total glucosides of peony may play an antidepressant effect by activating the hypothalamic–pituitary–adrenal axis. At the same time, the total glucosides of peony inhibit the decrease in monoamine oxidase activity and reduce its concentration in the brain, which has a certain effect on relieving depressive symptoms. The total glucosides of peony can also increase the expression of the nerve growth factor in the brain tissue of model rats [91]. With the gradual elucidation of the etiology and the pathogenesis of depression, the NO/cyclic guanosine phosphate (cGMP) pathway has also become one of the pathogeneses of depression and has attracted much attention [92]. The latest research [93] found that paeoniflorin can down-regulate the NO/c GMP pathway and play an antidepressant role.

### 4.7. Radix Glycyrrhizae

Radix Glycyrrhizae has the effect of tonifying the spleen and lungs, alleviating pain, and allowing the combination of various drugs. Modern studies have shown that Radix Glycyrrhizae has pharmacological effects, such as anti-inflammatory, antivirus, antitumor, and immune regulation effects [94]. The chemical components of Radix Glycyrrhizae are mainly flavonoids, saponins, polysaccharides, and coumarin compounds [95]. Flavonoids are one of the main chemical components of Radix Glycyrrhizae extract and are also important components reflecting the main medicinal value of Radix Glycyrrhizae, including dihydroflavonoids, dihydroflavonols, chalcones, isoflavanes, isoflavones, flavonoids, flavonols, isoflavones, and isoflavones [96]. Triterpene saponins include glycyrrhizic acid, glycyrrhetinic acid, glycyrrhizolide, and isoglycyrrhizolide [97].

Flavonoids are the most common components in Chinese herbs and have a broad spectrum of pharmacological activity. A recent study found that glycyrrhizin significantly reduced the symptoms of PSD, and its mechanism of action may be related to reducing the prefrontal cortex proapoptotic factor Bax expression and/or promoting the antiapoptotic factors Bcl-2 and BDNF expression, inhibiting amygdala apoptosis and regulating the expression of apoptosis-related factors in PSD rats [98,99,100]. In addition, studies have found that the total flavonoids of licorice can play an antidepressant role by anti-lipid peroxidation and reducing malondialdehyde production [101].

In summary, CHSG is involved in the treatment of PSD through different active ingredients, such as saikosaponin, hesperidin, paeoniflorin, total paeoniflorin, glycyrrhizin, the total flavonoids of licorice, etc. CHSG exerts anti-PSD effects as a result of the synergistic effects of multiple ingredients. However, the above studies still only scratch the surface of this topic and have not progressed to a deeper level, which awaits further study and development by researchers.

## 5. Summary

PSD is a common complication of psychosomatic disorder post-stroke, which not only affects the recovery of neurological function and reduces the quality of life, but also increases the disability rate and mortality [1]. At present, anti-PSD drugs, such as 5-HT and NE reuptake inhibitors, have a long drug course and many adverse effects. Therefore, the discovery of herbal formulas with definite efficacy, fewer toxic side effects, and less economic pressure in clinical observation and experimental research is an important way forward for traditional Chinese medicine.

CHSG is an effective formula for the treatment of PSD [102], the utility of which has long been proven, not only in centuries of clinical practice but increasingly also in experimental studies; however, the mechanism of action of CHSG in the treatment of PSD has still not been studied in depth and more detailed studies are lacking, and these problems have always restricted the development of TCM. Therefore, in future clinical and experimental studies, clinicians should pay attention to the changes in patients’ evidence patterns and provide more effective treatment modalities for patients.

## Figures and Tables

**Table 1 medicina-59-00055-t001:** Composition of CHSG.

Botanical Name	English Name	Part Used	Proportion
*Bupleurum chinese* DC.	Radix Bupleuri	Root	4
*Citrus reticulate* Blanco.	Pericarpium Citri Reticulatae	Pericarp	4
*Ligusticurn chuanxiong* Hort.	Rhizoma Chuanxiong	Rhizome	3
*Cyperus rotundus* L.	Rhizoma Cyperi	Rhizome	3
*Citrus aurantium* L.	Fructus Aurantii	Fruit	3
*Paeonia lactiflora* Pall.	Radix Paeoniae Alba	Root	3
*Glycyrrhiza uralensis* Fisch.	Radix Glycyrrhizae	Root and rhizome	1

**Table 2 medicina-59-00055-t002:** Clinical applications of CHSG in the treatment of PSD.

	Group	Medication (Dosage of Drug)	Number of People	Age (Average Age)	Treatment Effect	The Medication Time	Inclusion Time	Site	Reference
1	Observation group	Modified CHSG was applied on the basis of the control group: Rhizoma Chuanxiong, Radix Paeoniae Alba, Pericarpium Citri Reticulatae, Radix Bupleuri, Fructus Aurantii, Rhizoma Cyperi, Radix Glycyrrhizae: (10, 15, 6, 15, 10, 10, 6 g)	40	53–75	Prior treatment: HAMD (18.32 ± 1.36), PSQI (18.32 ± 1.35), NIHSS (7.23 ± 0.79), TCM syndrome score (22.21 ± 2.73) After treatment: HAMD (8.21 ± 1.03), PSQI (7.23 ± 0.95), NIHSS (2.15 ± 0.41), TCM syndrome score (11.05 ± 2.0)	CHSG: One dose per day, divided into two doses, for 8 weeks.	From August 2018 to August 2020.	Guangxi Veterans Hospital and Ruikang Hospital Affiliated to Guangxi University of Traditional Chinese Medicine	[10]
	Control group	Citalopram Hydrobromid	39	55–76	Prior treatment: HAMD (18.65 ± 1.38), PSQI (18.62 ± 1.32), NIHSS (7.52 ± 0.71), TCM syndrome score (23.22 ± 2.65) After treatment: HAMD (10.65 ± 1.09), PSQI (9.65 ± 0.85), NIHSS (3.59 ± 0.39), TCM syndrome score (14.22 ± 2.2).	Citalopram Hydrobromide: once a day, for 8 weeks.	From August 2018 to August 2020.	Guangxi Veterans Hospital and Ruikang Hospital Affiliated to Guangxi University of Traditional Chinese Medicine	[10]
2	Observation group	Modified CHSG was applied on the basis of the control group: Radix Bupleuri, Rhizoma Cyperi, Rhizoma Acori graminei, Radix Paeoniae Alba, Fructus Aurantii, Rhizoma Chuanxiong, Radix Glycyrrhizae (20,20,15,15,15,15,10 g)	41	59.8 ± 3.7	Prior treatment: HAMD (26.5 ± 4.1), NIHSS (12.1 ± 2.3). After treatment: HAMD (10.3 ± 2.6), NIHSS (5.0 ± 3.1). Total effective rate: 95.12%. Adverse reaction rate: 7.32%.	CHSG: One dose per day, divided into two doses, for 8 weeks.	From June 2016 to June 2017.	Wen County people’s Hospital	[11]
	Control group	Deanxit	41	59.9 ± 3.9	Prior treatment: HAMD (26.5 ± 4.1), NIHSS (12.2 ± 2.2). After treatment: HAMD (13.7 ± 4.3), NIHSS (7.2 ± 2.4). Total effective rate: 80.49%. Adverse reaction rate: 17.07%.	Twice a day, once in the morning and once in the evening for 4 weeks. It was given once every morning for another 4 weeks.	From June 2016 to June 2017.	Wen County people’s Hospital	[11]
3	Observation group	Modified CHSG was applied on the basis of the control group: Radix Bupleuri, Rhizoma Gastrodiae, Rhizoma Cyperi, Rhizoma Chuanxiong, Fructus Aurantii, Pericarpium Citri Reticulatae, Shao yao, Radix Glycyrrhizae (15,10,10,6,6,6,15,6 g)	30	52.23 ± 9.90	Prior treatment: HAMD (22.5 ± 4.48), NIHSS (9.33 ± 1.92), Serum hc-CRP (7.72 ± 1.18). After treatment: HAMD (8.67 ± 4.97), NIHSS (4.87 ± 1.41), Serum hc-CRP (4.42 ± 0.72). Total effective rate: 93%.	CHSG: One dose per day, divided into two doses, for 6 weeks.	From August 2014 to January 2016.	Affiliated Hospital of Nanjing University of Chinese Medicine	[12]
	Control group	Escitalopram Oxalate Tablets	30	50.73 ± 10.52	Prior treatment: HAMD (23.77 ± 4.63), NIHSS (9.50 ± 2.11), Serum hc-CRP (7.46 ± 1.35). After treatment: HAMD (12.40 ± 6.97), NIHSS (5.53 ± 1.33), Serum hc-CRP (5.12 ± 1.12). Total effective rate: 83%.	Once a day for 6 weeks.	From August 2014 to January 2016.	Affiliated Hospital of Nanjing University of Chinese Medicine	[12]

**Table 3 medicina-59-00055-t003:** Antidepressant effect and mechanism of CHSG.

Drug Composition (Dosage of Drug)	Dosage Forms and Effective Concentrations	Duration of Administration	Real Modules (Animal/Cell/Patients)	Possible Mechanisms	Apply Styles	Reference
CHSG: Radix Bupleuri, Fructus Aurantii, Radix Paeoniae Alba, Rhizoma Chuanxiong, Radix Glycyrrhizae, Rhizoma Cyperi, Pericarpium Citri Reticulatae (9, 9, 15, 9, 5, 9, 9 g).	water decoction; 5 g/kg, 10 g/kg.	14 days.	SD rat.	The contents of monoamine neurotransmitters 5-HT, NE and DA in hippocampus were increased.	In vivo.	[21]
Modified CHSG: Pericarpium Citri Reticulatae, Radix Bupleuri, Rhizoma Chuanxiong, Rhizoma Cyperi, Fructus Aurantii, Radix Paeoniae Alba, Radix Glycyrrhizae (Each dosage ratio was 10, 6, 6, 6, 6, 9, 3) + transcranial electric stimulation	water decoction; One dose was taken daily, and about 250 mL of juice was taken from each dose, divided into 2 packets, and taken in the morning and evening.	5 weeks.	Patient.	To increase the levels of 5-HT and BDNF in serum of patients.	In vivo.	[22]
CHSG: Radix Bupleuri, Radix Paeoniae Alba, Rhizoma Chuanxiong, Fructus Aurantii, Pericarpium Citri Reticulatae, Radix Glycyrrhizae, Rhizoma Cyperi (Each dosage ratio was 6:4.5:6:6:6:1.5:4.5).	water decoction; 5.9g/kg, 11.8 g/kg	21 days.	SD rat.	Regulation of BDNF/TrkB signaling pathway; It also reduced the expression of neuroinflammatory factors IL-6 and TNF-α.	In vivo.	[30]
CHSG: Radix Bupleuri, Pericarpium Citri Reticulatae, Rhizoma Chuanxiong, Rhizoma Cyperi, Fructus Aurantii, Radix Paeoniae Alba, Radix Glycyrrhizae (Each dosage ratio was 4:4:3:3:3:3:1).	water decoction; 5.9 g/kg, 11.8 g/kg.	4 weeks.	SD rat.	It increased the expression of BDNF and mBDNF.	In vivo.	[31]
CHSG: Radix Bupleuri, Pericarpium Citri Reticulatae, Rhizoma Chuanxiong, Rhizoma Cyperi, Fructus Aurantii, Radix Paeoniae Alba, Radix Glycyrrhizae (Each dosage ratio was 4:4:3:3:3:3:1)	water decoction; 5.9 g/kg	2 weeks.	SD rat.	Increased BDNF expression and TrkB mRNA expression in hippocampus, amygdala and frontal lobe.	In vivo.	[32]
CHSG: Radix Bupleuri, Fructus Aurantii, Radix Paeoniae Alba, Rhizoma Chuanxiong, Radix Glycyrrhizae, Rhizoma Cyperi, Pericarpium Citri Reticulatae (Each dosage ratio was 6:5:5:5:3:5:6)	medicinal slices; 1.4 g/kg	21 days.	SD rat.	Inhibition of neuronal apoptosis in hippocampal CA3 region.	In vivo.	[36]
CHSG: Radix Bupleuri, Pericarpium Citri Reticulatae, Rhizoma Chuanxiong, Rhizoma Cyperi, Fructus Aurantii, Radix Paeoniae Alba, Radix Glycyrrhizae (6, 6, 4.5, 4.5, 4.5, 4.5, 1.5 g).	water decoction; 7.875 g/kg.	21 days.	SD rat.	It can regulate the expression of Bcl-xs and Bcl-xl genes in hippocampus and inhibit the apoptosis of hippocampal neurons.	In vivo.	[39]
CHSG: Radix Bupleuri, Radix Paeoniae Alba, Fructus Aurantii, Rhizoma Chuanxiong, Pericarpium Citri Reticulatae, Rhizoma Cyperi, Radix Glycyrrhizae (6, 4.5, 4.5, 4.5, 6, 4.5, 1.5 g)	medicinal slices; 1 g/100 g.	21 days.	SD rat.	Autophagy was inhibited by reducing the protein expression levels of LC-3II/LC-3I and Beclin-1.	In vivo.	[42]
Modified CHSG: Radix Bupleuri, Angelica sinensis, Rhizoma Cyperi, Rhizoma Chuanxiong, Radix Curcumae, Rhizoma Atractylodis Macrocephalae, Fructus Aurantii, Pericarpium Citri Reticulatae, Radix Paeoniae Alba, Radix Glycyrrhizae (15, 15, 10, 10, 10, 10, 10, 10, 10, 6 g).	water decoction; One dose was taken daily, and about 300 mL of juice was taken from each dose, divided into 2 packets, and taken in the morning and evening.	8 weeks.	Patient.	The levels of serum inflammatory factors (IL-6, CRP, TNF-α) were reduced and the inflammatory response was reduced.	In vivo.	[46]

## Data Availability

The data are available upon request from the author.

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
