# Peer review of "A Chinese Classical Prescription Chaihu Shugan Powder in Treatment of Post-Stroke Depression: An Overview"

_medicina, 2022, doi:10.3390/medicina59010055_

Round 1
Reviewer 1 Report
The feedbacks:
1. Introduction
a. Please describe the meaning of ‘simple’ stroke patients in the following statement in the Introduction ‘…Some studies have found that the mortality rate of PSD is about 1.28–1.75 times that of simple stroke patients [3].’?
b. Please explain the meaning of ‘qi’.
2. Section 2.1
a. Is it quite confusing when reading this statement, 'After 45 days of CHSG treatment, PSD patients have no obvious depression symptoms such as loss of interest and depression?' Why depression is depression (depressive) symptoms?
b. Regarding Professor Sheng's experience in PSD treatment (reference number 7), did the improvement (HAM-D, HAMA, and SDS scores) observe solely due to CHSG treatment? Did the professor also use repetitive transcranial magnetic stimulation? If so, please also include this in the statement.
c. There is a repetition of the statement in this sentence ‘He divided PSD into the type of liver stagnation and spleen deficiency, the type of mutual obstruction of phlegm and blood stasis, and the type of mutual obstruction of phlegm and blood stasis [8].’ Please clarify.
d. Repetition is the word ‘depression’ in the following statement ‘ the patient's family members complained that the patient's depression and depression were significantly improved…’.
3. Section 2.2
a. What is Delixin? Did you mean deanxit?
b. The use of rapidly in the following statement might be tricky or inaccurate ‘…with Delixin could rapidly reduce the HAMD score..’. How did you define rapidly in the statement? Are two months considered rapid?
c. Please elaborate more on the outcomes of HAMD, NIHSS and serum hs-CRP for CHSG + escitalopram vs escitalopram alone. Any difference between these two groups?
d. What is CHSGS? Did you mean CHSG?
e. What is basic stroke therapy? Please include a brief description of the term.
f. Did the HAM-D score really increase in the study conducted by Zhou et al. (Reference no. 13)? The increase in HAM-D score typically correlates with higher depressive symptoms.
4. Section 2.3:
a. Describe the side effects (examples of side effects) found in the studies in section 2.3 (Reference no 14, 15 and 16). Also, please elaborate more on which side effects are less common with CHSG compared to standard treatment with antidepressants.
b. Reference no. 16. Interestingly, in the statement, combined CHSG and citalopram were better than citalopram alone. Please give more details about the side effects between CHSG + citalopram and the citalopram alone group. If different, state the potential mechanism of reduced adverse effects in one group compared to another.
c. Does the statement ‘CHSG reduce the occurrence of adverse effects of PSD’ accurate? What is the adverse effect of PSD? Do you mean the adverse effects of PSD treatment? If referring to PSD treatment, please indicate what kind of treatment.
5. Section 3.1
a. Describe in full words the term BDNF before the use of the abbreviation.
b. Explain how the increase in BDNF and TrKB is related to the monoamine hypothesis (particularly neurotransmitter) in depression. There is a missing link in this paragraph.
c. Briefly explain the function of Bcl-xs and Bcl-xl.
d. Please describe why the effects of CHSG on the hippocampal CA3 region are related to the anti-depressive effect.
e. Describe the function of LC-3II/I and Beclin-1 in apoptosis.
6. Section 3.3
a. Please recheck the description of the HPA axis. Is H referred to as the hypothalamus or thalamus?
7. Section 4.1
a. Explain how inhibition of acetylcholinesterase expression and activity are related to the anti-depressive effects of CHSG.
8. Section 4.2
a. Elaborate more on the anti-depressive effects of Rhizoma Cyperi extracted with ethanol (Reference no 42).
9. Section 4.3
a. Describe in full words the term CREB before the use of the abbreviation.
10. Section 4.6
a. Repetition of paeoniflorin word.
Author Response
请参阅附件。

Reviewer 2 Report
Authors in the review A Chinese classical prescription Chaihu Shugan Powder in treatment of Post-stroke depression: An overview summarize available data from clinical and preclinical studies, concerning Chaihu Shugan Powder, traditional chinese medicine, and its possible use in the treatment of post-stroke depression. However, there are some discrepancies and inaccuracies in the text that should be addressed.
1. Fig.1 and Table 1 show the same thing, so I think they can be merged into 1 fig./table to avoid duplication
2. In the whole text there is no mention about the treatment protocol (i.e. how often and how long should be the CHSG applicated) and the form of CHSG application (pills, tea, ....)
3. I think that in some chapters authors should provide data from more studies to increase credibility of the reviewed treatment; for example in chapter 2.3 (they can include data from study PMID: 33657988) or in chapter 3.1. (data from study 31630361) etc.
4. In chapter 4.2 authors mentioned that ....“Rhizoma Cyperi were found to have similar potency to the control fluoxetine, both of which shortened the swimming time and the immobility of the hanging tail......, confirming the significant antidepressant effect....“
I think that shortening of the swimming time does not support its antidepressant properties, it means exactly the opposite.
5. Many of the data sources of the manuscript (i.e. journals) are not indexed in the Scopus or Web of Science databases. I believe that incorporation of the data from the internationally recognized journals would increase the credibility of the manuscript.
6. English language should be carefully checked.
Round 2
Reviewer 1 Report
The authors have addressed all my feedback and improved the article significantly. I have no further questions.